# Regional prevalence of hypertension among people diagnosed with diabetes in Africa, a systematic review and meta-analysis

Thomas Hinneh[1]*, Samuel Akyirem[2], Irene Fosuhemaa Bossman[3], Victor Lambongang[4], Patriot Ofori-Aning[5], Oluwabunmi Ogungbe[1,6], Yvonne Commodore Mensah[1,6]

1 Johns Hopkins University School of Nursing, Baltimore, Maryland, United States of America, 2 Yale School of Nursing, Yale University, New Haven, Connecticut, United States of America, 3 School of Health and Life Sciences, Glasgow Caledonian University, Scotland, United Kingdom, 4 School of Health Sciences, Liberty University, Lynchburg, Virginia, United States of America, 5 Department of Medicine for Older People, Stockport NHS Foundation Trust, Manchester, United Kingdom, 6 Johns Hopkins Bloomberg School of Public Health, Baltimore, Maryland, United States of America

* thinneh1@jh.edu

**Data Availability Statement:** All data used in this manuscript has been included in this manuscript.

**Funding:** The authors received no specific funding for this work.

## Abstract

Hypertension and diabetes comorbidity can increase healthcare expenditure and the risk of coronary heart disease. We conducted a systematic review and meta-analysis to estimate the prevalence of hypertension among people with diabetes in African countries. We searched EMBASE, PubMed and HINARI databases from inception to March 2023. Cross-sectional studies reporting the prevalence of hypertension among people with diabetes and published in English in Africa were eligible for inclusion. The cross-sectional study design component of the mixed method appraisal tool was used to assess the quality of the included studies. We quantified the overall and regional prevalence of hypertension among people with diabetes using random-effects meta-analysis. We assessed heterogeneity and publication bias using $I^2$ statistics and funnel plots. Out of 3815 articles retrieved from the various databases, 41 met the inclusion criteria with sample sizes ranging from 80 to 116726. The mean age was 58 years (± 11) and 56% were women. The pooled prevalence of hypertension in people diagnosed with diabetes was 58.1% [95% CI: 52.0% - 63.2%]. By region, Central Africa had the highest hypertension prevalence; 77.6% [95% CI: 53.0% - 91.4%], Southern Africa 69.1% [95% CI: 59.8% - 77.1%;], North Africa 63.4% [95% CI: 37.1% - 69.1%;], West Africa 51.5% [95% CI: 41.8% - 61.1%] and East Africa 53.0% [95% CI: 45.8% - 59.1%]. Increasing age, being overweight/obese, being employed, longer duration of diabetes, urban residence, and male sex were reported to be associated with a higher likelihood of developing hypertension. The high prevalence of hypertension among people with diabetes in Africa highlights the critical need for an integrated differentiated service delivery to improve and strengthen primary care and prevent cardiovascular disease. Findings from this meta-analysis may inform the delivery of interventions to prevent premature cardiovascular disease deaths among persons in the region.

**Competing interests:** The authors have declared that no competing interests exist.

## Introduction

Hypertension is the most common leading cause of cardiovascular diseases (CVDs), and a major cause of morbidity and mortality globally [1]. The prevalence of hypertension has risen substantially to over one billion since 1990, according to the Lancet NCD Risk Factor Collaboration Study (NCD-RisC) [1]. During the same period (1990–2017), the global burden of diabetes also increased from 211.2 million to 476 million [2]. Nearly two-thirds of hypertension and diabetes cases occur in low- and middle-income countries (LMICs), where health systems are already burdened with infectious diseases [3]. Even though the Lancet NCD-RisC study reported a decline in the burden of hypertension in high-income countries, LMICs continue to experience an upsurge in rates of hypertension [2–4].

Diabetes and hypertension have a higher likelihood of co-occurring due to the commonalities of risk factors including low physical activity, unhealthy dietary patterns, and obesity [1].

Hypertension and diabetes are linked pathophysiologically. Diabetes can cause microvascular changes including the stiffening of small blood vessels (a condition known as arteriosclerosis). Arteriosclerosis can also elevate peripheral vascular resistance and can significantly increase the risk of developing hypertension [5]. Apart from neuropathic, nephropathic disorders and CVDs, hypertension is the most common comorbid condition and the leading cause of death among people with diabetes [6]. Although the prevalence of hypertension is well studied in sub-Saharan Africa, there is limited research on the prevalence of hypertension among this population who experience a greater burden of hypertension-associated mortality [1].

Comorbid hypertension worsens blood pressure control and quality of life of patients living with diabetes [7–9], increases healthcare costs and complicates healthcare needs. Given the increase in healthcare needs of the diabetes population as a result of the comorbidity, such patients are less likely to receive appropriate care until the occurrence of complications [10]. These negative outcomes of hypertension and diabetes comorbidity underscore the need for an equal prioritization of clinical and community-based interventions for hypertension and diabetes. Moreover, a knowledge of the prevalence of hypertension among healthcare providers and patients is critical to enhancing screening and early case detection and management strategies within the clinical settings to avert negative outcomes from both hypertension and diabetes.

Estimating the prevalence of hypertension among people with diabetes will inform health systems priorities and health policies for CVD response in the African region in reference to achieving a 25% decrease in hypertension by the end of 2025 [11]. To this end, we aimed at assessing prevalence estimates of hypertension among patients with diabetes in Africa.

## Methods

This study was conducted according to a pre-designed protocol and the Preferred Reporting Items for a Systematic Review and Meta-analysis (PRISMA) guideline (**S1 Checklist**) [12]. The review was registered on the International Prospective Register of Systematic Reviews (Prospero Registration ID: CRD42021256221) [13].

### Search strategy

A comprehensive literature search was performed on PubMed, EMBASE, and HINARI databases to identify relevant articles that provided data on the prevalence or incidence of hypertension among people diagnosed with diabetes in Africa. The search terms were categorized into population, outcome/phenomenon of interest, and context In accordance with the recommendation of the Joanna Briggs Institute (JBI) Manual for Evidence Synthesis approach for drafting review questions [14]. Specific terms developed under each category were (1)

population: adults diagnosed with type 1 and type 2 diabetes mellitus, with or without hypertension, (2) Outcome/phenomenon of interest: prevalence, incidence, risk factors, co-morbidity, treatment outcomes of diabetes and/or hypertension, and (3) Context: Africa. Based on these pre-specified search terms, a search string for: "Diabetes type 1" OR "Diabetes type 2" AND "hypertension" OR "high blood pressure" AND "Incidence" OR "prevalence" OR "co-morbidity" OR "risk factors" OR "treatment outcomes" AND "Africa" OR "All African countries" was derived through an iterative process and adapted for all the databases. Medical Subject Headings (MeSH) were applied for terms in PubMed. The original search covered from inception to September 2021 on each database. An updated search was conducted in March 2023, to cover the entire period from inception to March 2023. This review is the first study to estimate the prevalence of hypertension among people with diabetes, so no limitation was applied in terms of the year of publication. The full search strategy is available (**S1 Table**).

## Eligibility criteria

We used the "population, outcome/phenomenon of interest, and context" strategy to guide the inclusion and exclusion criteria. The population included participants diagnosed with type 1 diabetes mellitus (T1DM) or type 2 diabetes mellitus (T2DM). Only studies of cross-sectional design were included. The outcome/phenomenon of interest was the prevalence of hypertension among people with diabetes. We excluded studies that included pregnancy-induced hypertension as an outcome. Hypertension threshold was specified as ≥140/90 mm Hg or ≥130/80 mm Hg according to the World Health Organization guideline [15]. Only studies conducted in African countries were included in the review.

## Study selection and data extraction

Articles retrieved from the various databases were exported to Endnote Version X 9.0 software and duplicates were removed. Two reviewers, IB and VL independently conducted title and abstract screening using the Rayyan Software platform. TH and SA resolved any disagreements when the two reviewers could not reach a consensus. TH, OP, and SA independently extracted the data from the included articles using Microsoft Office Excel. Data items including study title, authors, date of publication, region, country, type of participants, age, prevalence of hypertension, complications, diagnostic thresholds for hypertension and diabetes, and factors associated with hypertension among people with diabetes were extracted.

## Risk of bias assessment

TH and SA independently conducted quality assessments using a Mixed-Method Appraisal Tool (MMAT) [16]. We initially applied the first two mandatory questions to ascertain the feasibility of the tool. We used the standardized set of questions, Q4.1-Q4.5 which is designed for cross-sectional study designs. The overall quality score of the studies was the average of the independent scores of the two reviewers. Studies that met 4 or 5 of the quality assessment criteria were adjudged to be of high quality and low risk of bias. A rating of 3 means the study met three of the quality assessment criteria, and medium quality and medium risk of bias. However, studies rated 2 or 1 showed poor quality and had a high risk of bias.

## Statistical analysis

We performed a tabular synthesis of sample characteristics, effect measures (prevalence), main findings, diagnostic threshold, and criteria of included studies. The prevalence of hypertension among people diagnosed with diabetes was pooled in a random-effects meta-analysis using the

"metafor" package in R. We used DerSimonian-Laird's random-effects model to determine the pooled prevalence as we anticipated considerable heterogeneity among the various studies [17]. We also computed 95% confidence interval (CI) for individual studies and the pooled prevalence using the Clopper-Pearson interval. Furthermore, we used narrative synthesis to summarize the factors associated with hypertension among people with diabetes reported by the studies. Heterogeneity among studies was assessed using $I^2$ statistic and with a 10% level of statistical significance [18]. No cut-off was set for heterogeneity. However, we performed a sub-group analysis based on regional blocks in Africa (East, West, North, South, and Central), diabetes sub-types, and risk of bias scores. We assessed publication bias using funnel plots and Peter's test for funnel plot asymmetry.

## Results

### Study results and identification

The initial electronic search yielded 3807 records. De-duplication led to the exclusion of 564 articles; the 3243 articles progressed through titles and abstracts screening, with the exclusion of a further 3126 records. The full texts of the remaining 117 records were extensively reviewed and guided by the eligibility criteria. Three additional articles were identified through hand searching of reference lists. The updated search yielded an additional five studies, published between 2021–2022. Overall, 41 articles fully met our inclusion criteria and were included in this review and the meta-analyses. Details of the screening process are provided in the PRISMA flowchart in (**Fig 1**).

### Characteristics of included studies

Studies were representative of the five regions in Africa. There were more studies conducted in East (n = 15) [19–34], West (n = 10) [35–44], North (n = 4) [45–48], Central (n = 2) [49, 50] and South Africa (n = 10) [51–60]. The included studies were published between 2002 and 2022 and the summary of findings is shown in **Table 1**. The sample sizes of the studies ranged from 80–116726 and the mean age of the participants was 58 years (± 11) (range: 15–87).

While most 33 (80%) of the studies utilized a cut-off point of ≥140/90 mm Hg for hypertension diagnosis, some 4 (10%) used ≥130/80 mm Hg, and the remaining 4(10%) did not report. A few of the studies included in the review used a hemoglobin A1c (HbA1c) threshold between (≥5.7 and >7.0) [20–23, 26–29, 31, 34, 35, 37, 39, 46, 48, 50, 53, 58, 59]. In some cases, a fasting plasma/blood glucose level of 5–7 mmol/L or more was used [61]. Other studies used HbA1c levels exceeding 6.5 or 7 as a diagnostic criterion for diabetes [20, 21, 31, 34, 43, 46, 57–59]. Twenty-four studies (59%) included patients diagnosed with only T2DM, seventeen (41%) included both T1DM and T2DM and no study was found among only patients with T1DM. Overall, the prevalence of hypertension among people with diabetes ranged from 21% in a study conducted in Ghana [37] to 92% in South Africa [58].

### Prevalence of hypertension in people with diabetes (T1DM and T2DM)

Forty-one studies had complete data on the prevalence of hypertension among people diagnosed with diabetes and hence were eligible for inclusion in the meta-analysis. Overall, the prevalence of hypertension among people diagnosed with diabetes was 58.1% (95% CI: 52.%–63.2%) shown in (**Fig 2**). This is further depicted in a heat map of Africa highlighting the regional prevalence of hypertension among people with diabetes (**Fig 3**). There was high and significant statistical heterogeneity across studies ($i^2$ = 98%, p<0.001). The prevalence of hypertension among people diagnosed with diabetes within the five geopolitical regions; East

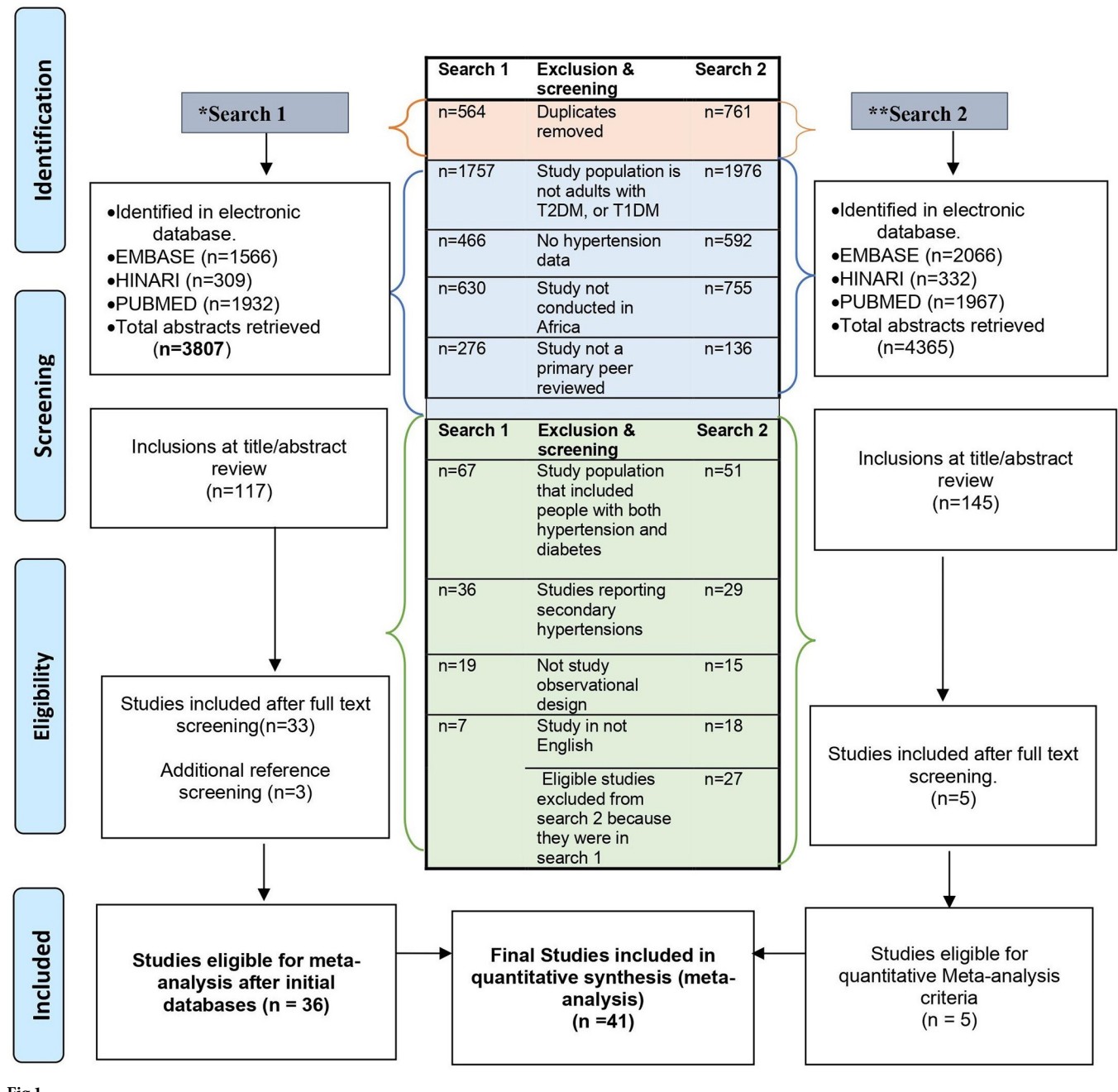

**Fig 1.**

Africa 53.0% [95% CI: 45.8%–59.1%] (**S1 Fig**), West Africa 51.5% [95% CI: 41.8%–61.1%] (**S2 Fig**), North Africa 63.4% [95% CI: 37.1%–69.1%;] (**S3 Fig**), South Africa 69.1% [95% CI: 59.8%–77.1%;] (**S4 Fig**), and with Central Africa had the highest hypertension prevalence; 77.6% [95% CI: 53.0%–91.4%] (**S5 Fig**).

## Prevalence of hypertension in patients with only T2DM

Twenty-four studies (59%) included people diagnosed with T2DM (**Fig 4**). Sub-group analysis among those with T2DM showed a prevalence of 60.8% (CI:53.4%-67.7%; $i^2$ = 99%, p<0.01)

**Table 1. Studies included that shows the prevalence of hypertension among people with diabetes in Africa, 2002–2022.**

| Citation | Country | Study design | Sample size | Sample characteristics | Criteria for DM diagnosis | Criteria for HTN diagnosis | Major findings |
|---|---|---|---|---|---|---|---|
| Abdelbagi et al (2021) | Sudan | Cross-sectional | 1973 (1155 females) | Median age– 58 years, range 50 to 65 years. Both T1DM and T2DM participants included | HbA1c > 6.5% | ≥ 140/ 90 mmHg | 47.6% of people with diabetes had HTN. Being male, employed, obese and old are associated with HTN in patients with DM |
| Berraho et al (2012) | Morocco | Cross-sectional | 525 (361 females) | No Median or mean age of participants presented. Only T2DM patients included | Not provided | ≥ 140/ 90 mmHg | 70.4% of people with diabetes had HTN. Associated factors include age, BMI and duration of diabetes |
| Choukem et al (2007) | Cameroon | Cross-sectional | 210 (104 females) | Age, Mean = 56.6 years (15–86years). T1DM and T2DM patients included | FCG >126 mg/dl | ≥130/80mm/hg | 66.70% of people with diabetes had HTN. Associated factors include BMI, obese and duration of diabetes |
| Dagnew et al (2019) | Ethiopia | Cross-sectional | 315 (153 females) | Age, mean = 54.1 years, range (22-87years). Only T2DM patients are included. | FPG > 126 mg/dl | ≥140/90mm/hg | 41% of people with diabetes had HTN. Associated factors include males, old age, employed, higher education and being single |
| Demoz et al (2019) | Ethiopia | Cross-sectional | 357 (189 females) | Mean age = 56 years. Only T2DM patients were included | HbA1c<7% | Not reported | 52.70% of people with diabetes had HTN |
| Makwero et al (2018) | Lesotho | Cross-sectional | 150 (121 females) | Age, mean = 58.2years, range 19-97years. Both T1DM and T2DM participants included | HbA1c > 7.0% | >130/80 mmHg | 85.3% of people with diabetes had HTN |
| Mogre et al (2014) | Ghana | Cross sectional Study | 100 (77 females) | Mean age = 67.53 years. T2DM patients | FPG ≥6.1 mmol/L | ≥140/90mm/hg | 21% of people with diabetes had HTN |
| Regassa et al (2020) | Ethiopia | Cross-sectional Study | 454 (195 females) | Mean age = 45.39 years, range (15 – 86years) included only T2DM participants. | Not specified | >139/89 mmHg. | 60% of people with diabetes had HTN |
| Abdissa et al 2020 | Ethiopia | Cross-sectional Study | 366 (163 females) | Mean age = 50.1 years. Both T1DM and T2DM participants included | Not specified | ≥140/90 mmHg | 37.40% of people with diabetes had HTN. Associated factors include age, BMI, and Khat chewing |
| Adeniyi et al 2016 | South Africa | Cross sectional Study | 265 (190 females) | Mean age = 56.7 years. Included only T2DM participant | Not specified | ≥140/90 mmHg | 75.5% of people with diabetes had HTN. Associated factors include male, age, unemployed, excessive alcohol intake and consumption |
| Arije et al 2007 | Nigeria | Cross sectional Study | 256 (146 females) | Mean age = 59.1 years, range (21-83yrs) including only DM patients [type not specified]. | Not specified | 140/90mm/HG | 42.20% of people with diabetes had HTN. |
| Githinji et al 2018 | Kenya | Cross-sectional study | 1548 (919 females) | Mean age = 58 years. Both T1DM and T2DM participants included | Not specified | ≥140/90 mmHg | 34% of people with diabetes had HTN |
| Ndege et al 2014 | Ethiopia | Cross sectional Study | 218 (122 females) | Mean age = 57 years. Only T2DM participants included | Not specified | ≥140/90 mmHg | 85% of people with diabetes had HTN |
| Thinyane et al 2013 | Lesotho | Cross-sectional study | 80 (39 females) | Median age 49 with age range (36 0 56 years) Both T1DM and T2DM participants included | Not specified | ≥140/90 mmHg | 56% of people with diabetes had HTN |
| Mengesha et al 2007 | Botswana | Cross sectional Study | 401 (287 females) | Age range (30–70years) Both T1DM and T2DM participants included | Not specified | ≥140/90 mmHg | 61.20% of people with diabetes had HTN. Associated factors include age, sex, type of DM, BMI and hypertriglyceridemia |

(*Continued*)

**Table 1.** (Continued)

| Citation | Country | Study design | Sample size | Sample characteristics | Criteria for DM diagnosis | Criteria for HTN diagnosis | Major findings |
|---|---|---|---|---|---|---|---|
| Akalu et al 2020 | Ethiopia | Cross sectional study | 378 (153 females) | Mean age = 56 years. Only participants with T2DM included | HbA1c ≥6.5% | ≥140/90 mmHg | 59.50% of people with diabetes had HTN. Associated factors include age range (50 -60yrs), patients from urban area, BMI, longer duration of T2DM, patients with poor glycaemic control and patients who were current cigarette smokers |
| Awadalla et al 2017 | Sudan | Cross sectional study | 424 (209 females) | Age range (20 – 75years) T2DM patients included | HbA1c ≥7.1% | ≥140/90mm/Hg | 39.90% of people with diabetes had HTN. Associated factors including longer duration of diabetes and living in urban areas |
| Chetty et al 2021 | South Africa | Cross sectional | 4122 (3072 females) | Mean age = 59.21 years, range (45 – 74yrs); T2DM participants included | Not specified | Not reported | 77.90% of people with diabetes had HTN. Associated factors include female |
| Gezawa et al 2019 | Nigeria | Cross sectional study | 220 (123 females) | Age range (35–65 years) including T2DM and T1DM diabetic patients | Not specified | SBP≥140mmHg or DBP≥90mmHg | 46.80% of people with diabetes had HTN |
| Hussein et al 2020 | Egypt | Cross sectional study | 800 (200 females) | Age, mean = 58.2 years. T2DM patients included. | HbA1c ≥7.0% | ≥140/90 mmHg | 60.25% of people with diabetes had HTN |
| Kemche et al 2020 | Cameroon | Cross sectional study | 109 (66 females) | Mean age = 55 years. Both participants with T1DM and T2DM included | Not specified | ≥140/90mm/Hg | 86.20% of people with diabetes had HTN. Associated factors include eating more than 2 times a day and long duration of diabetes |
| Tsegaw et al 2021 | Ethiopia | Cross sectional Study | 739 (307 females) | Age range (26 – 85years) including T2DM participants | HbA1c >7.0–9.0% | 140/90mm/Hg | 50.20% people with diabetes have HTN |
| Unadike et al 2011 | Nigeria | Cross sectional Study | 450 (225 females) | Mean age = 51.1 years; Only participants with T2DM included. | Not specified | 140/90mm/Hg | 54.20% of people with diabetes had HTN |
| Wanjohi et al 2002 | Ethiopia | Cross Sectional Study | 100 (63 females) | Median age 53.7 years, Age range (34-80years) included Patients with T2DM only | HbA1c≥7.0 FPG ≥5.9mmol/L | Not reported | 50% of people with diabetes had HTN |
| Amankwah-Poku et al 2020 | Ghana | Cross-sectional study | 162 (128 Females) | Mean age = 61.0 years Recruited participants with T2DM | Not reported | >140/90mm/Hg | 79.6% people with diabetes had HTN |
| Danquah et al 2012 | Ghana | Cross-sectional study | 675 (323 Females) | Mean age,54.7 years; Recruited T2DM participants | FPG≥ 7mmol/L | ≥140/90mm/Hg | 63% of people diagnosed with diabetes had HTN. Family history of diabetes, working time, increased triglycerides were associated with HTN |
| Kalain & Omole 2020 | South Africa | Cross-sectional Study | 200 (128 Females) | Mean age,57.8 years; Recruited T2DM participants | HbA1c ≥7.0% | ≥140/90mm/Hg | 92% of people diagnosed with diabetes had HTN. |
| Goie & Naidoo 2016 | South Africa | Cross sectional Study | 280 (201 were female) | Mean Age 59 years, Recruited T2DM participants | Not Reported | ≥140/90mm/Hg | 57.5% of people diagnosed with diabetes had HTN. |
| Thomas et al 2013, | South Africa | Cross-sectional Study | 5565 (2016 Females) | Mean Age, 50.8 years, Recruited people with T1DM and T2DM participants | HbA1c ≥7.0% | ≥140/90mm/Hg | 44.0% of people diagnosed with diabetes had HTN. |
| Boake & Mash 2022 | South Africa | Cross-sectional Study | 116726 (74471 females) | Mean Age, 61.4 years, Recruited people with T1DM and T2DM participants | HbA1c ≥7.0% | ≥140/90mm/Hg | 69.5% of people diagnosed with diabetes had HTN. |

(*Continued*)

**Table 1.** (Continued)

| Citation | Country | Study design | Sample size | Sample characteristics | Criteria for DM diagnosis | Criteria for HTN diagnosis | Major findings |
|---|---|---|---|---|---|---|---|
| Rotchford & Rotchford 2002 | South Africa | Cross sectional Study | 253 (183 were females) | Mean Age, 56.5 years, Recruited people with T1DM and T2DM participants | HbA1c >5.7% | ≥140/90mm/Hg | 41.8% of people diagnosed with diabetes had HTN. |
| Mfeukeu-Kuate et al. 2022 | Cameroon | Cross-sectional study | 112(49 were females) | Median age was 58 years, recruited participants with T2DM | Not Reported | ≥140/90mm/Hg | 50.4% of people diagnosed with diabetes had HTN. |
| Amoussou-Guenou et al. 2015 | Benin | Cross-sectional study | 400 (264 were females) | Median age was mean age was 55.6 recruited participants with T2DM and | HbA1c > 7% | ≥140/90mm/Hg | 70.0% of people diagnosed with diabetes had HTN. Above 555 years is associated with hypertension |
| Ovono et al 2011 | Gabon | Cross-sectional study | 1300 (699 were females) | Mean age was 53 years, recruited participants with T2DM | HbA1c > 7% | ≥140/90mm/Hg | 35% of people diagnosed with diabetes had HTN |
| Dzudie et al 2012 | Cameroon | Cross-sectional study | 420 (215 were females) | Mean age was 55.9 years, recruited participants with T2DM | Not Reported | >140/90mm/Hg | 50.2% of people diagnosed with diabetes had HTN |
| Kimando et al 2017 | Kenya | Cross-sectional study | 385 (253 were females) | Mean age was 63.3 years, recruited participants with T2DM | HbA1c > 7% | ≥140/90mm/Hg | 49.6% of people diagnosed with diabetes had HTN. Old age above 50 yrs., longer duration with diabetes above 5 yrs. and advanced stages of CKD |
| Chahbi et al 2018 | Morrocco | Cross-sectional study | 300 (279 were females) | Mean age was 57.24 years recruited, participants with T2DM | HbA1c >7% | ≥140/90mm/Hg | 44.3% of people diagnosed with diabetes had HTN |
| Kilonzo et al 2017 | Tanzania | Cross-sectional study | 295 (161 were females) | Mean age was 57 years, recruited, participants with T2DM | FBG >7.0mmol/L. | ≥130/80mmHg | 69.8% of people diagnosed with diabetes had HTN |
| Munyogwa et al. 2020 | Tanzania. | Cross-sectional study | 330 (189 were females) | Mean age was 40.27 years, recruited, participants with T2DM | FBG ≥11.1 mmol/l | ≥140/90mm/Hg | 63.3% of people diagnosed with diabetes had HTN |
| Mwita et al 2019 | Tanzania | Cross-sectional study | 150 (93 were females) | Mean age was 51.6 years, recruited, participants with T2DM and T1DM | Not Reported | ≥140/90mm/Hg | 54.7% of people diagnosed with diabetes had HTN |
| Kahloun et al.2014 | Tunisia | Cross-sectional study | 2320 (1396 were females) | Mean age was 54.5 years, recruited, participants with T2DM | Not Reported | Not Reported | 37.5% of people diagnosed with diabetes had HTN |

SBP–Systolic Blood Pressure, DBP–Diastolic Blood Pressure, HTN–Hypertension, T1DM–Type 1 diabetes mellitus, T2DM–Type 2 diabetes mellitus, FBG–Fasting Blood Glucose; HbA1C –Hemoglobin A1C; FCG–Fasting Capillary Glucose; FPG–Fasting Plasma Glucose

which was higher than the overall prevalence among T1DM and T2DM "combined" (58%). The highest prevalence was in South Africa (92%) followed by Ethiopia (84.9%), Zimbabwe (80%), and Ghana (74.6%) among participants with only T2DM. Some countries had only one study, hence estimation of the lowest prevalence among people with T2DM was impossible.

## Assessment of risk of bias

A summary of the risk of bias is shown in (**S2 Table**). Thirty-six of the studies had a score of 4 and 5 suggesting high quality and low risk of bias. Most of the studies had adequate sample sizes and were representative enough. However, few studies were at risk of bias due to lack of sample representativeness [25, 26, 37, 45, 49, 50, 53–55, 62]. Statistical appropriateness for the research question was unclear or not consistent with the study aims [22, 28, 33, 36, 38, 41, 50, 54–56, 59]. Except for Abdelbagi, et al, the sampling strategy was clearly stated in all the studies

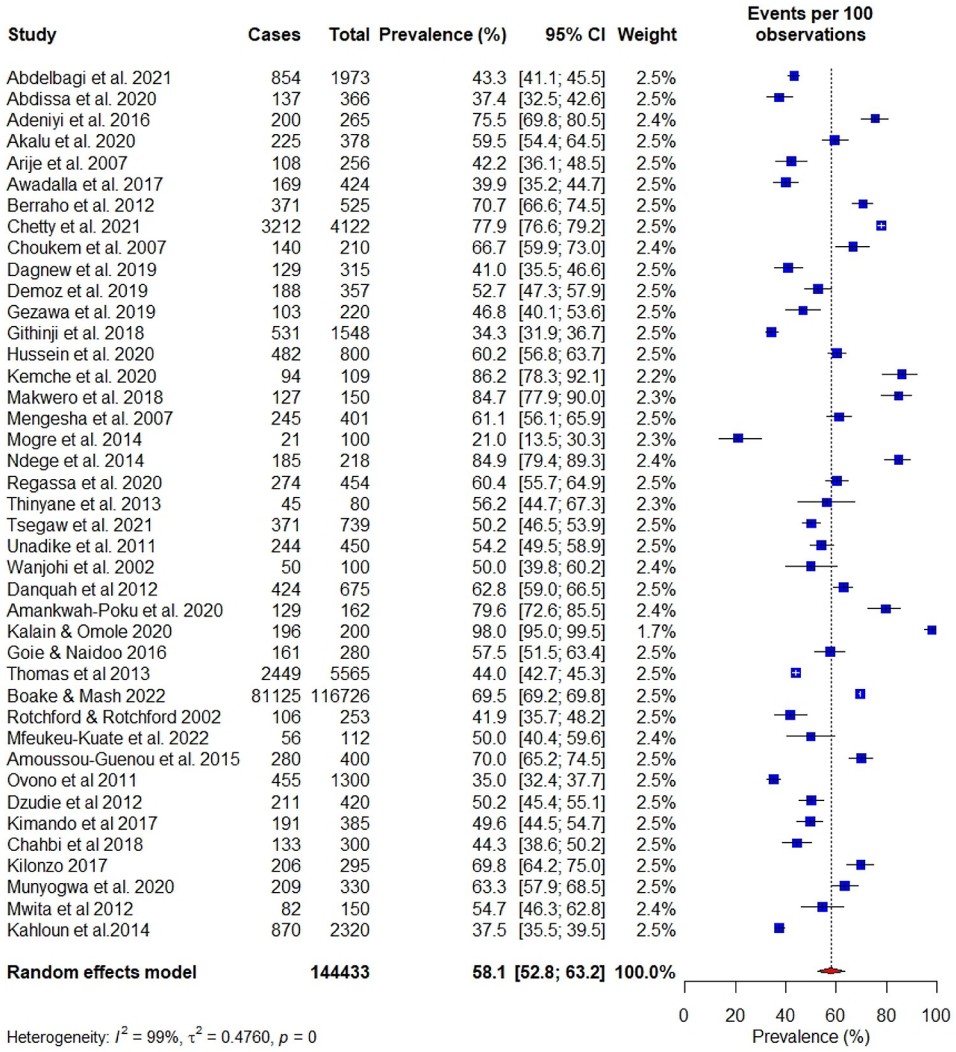

**Fig 2.**

[20]. Although no study was excluded based on the risk assessment score, we conducted further analysis using studies with scores 4 and 5 and found relatively same prevalence (**S6 Fig**). The prevalence estimates of hypertension according to the risk of bias score were relatively the same as the overall prevalence estimate for the overall studies and among studies that included only participants with type 2 diabetes (**S7 Fig**). The result of the linear regression test of funnel plot asymmetry; t = -2.22, df = 39, p-value = 0.0322, showed evidence of plot asymmetry, which is suggestive of publication bias (**Fig 5**).

## Factors associated with hypertension among people with diabetes (T1DM and T2DM)

Some studies reported factors associated with developing hypertension among people with diabetes. For instance, Abdellagi and colleagues found that the odds of developing hypertension increased with increasing age, body mass index, and being employed. In one study, age increased the odds of hypertension by more than 600% (OR 7.26 4.20–12.54) [45]. Furthermore, the risk for hypertension among people with diabetes is highest within the 50–60 age

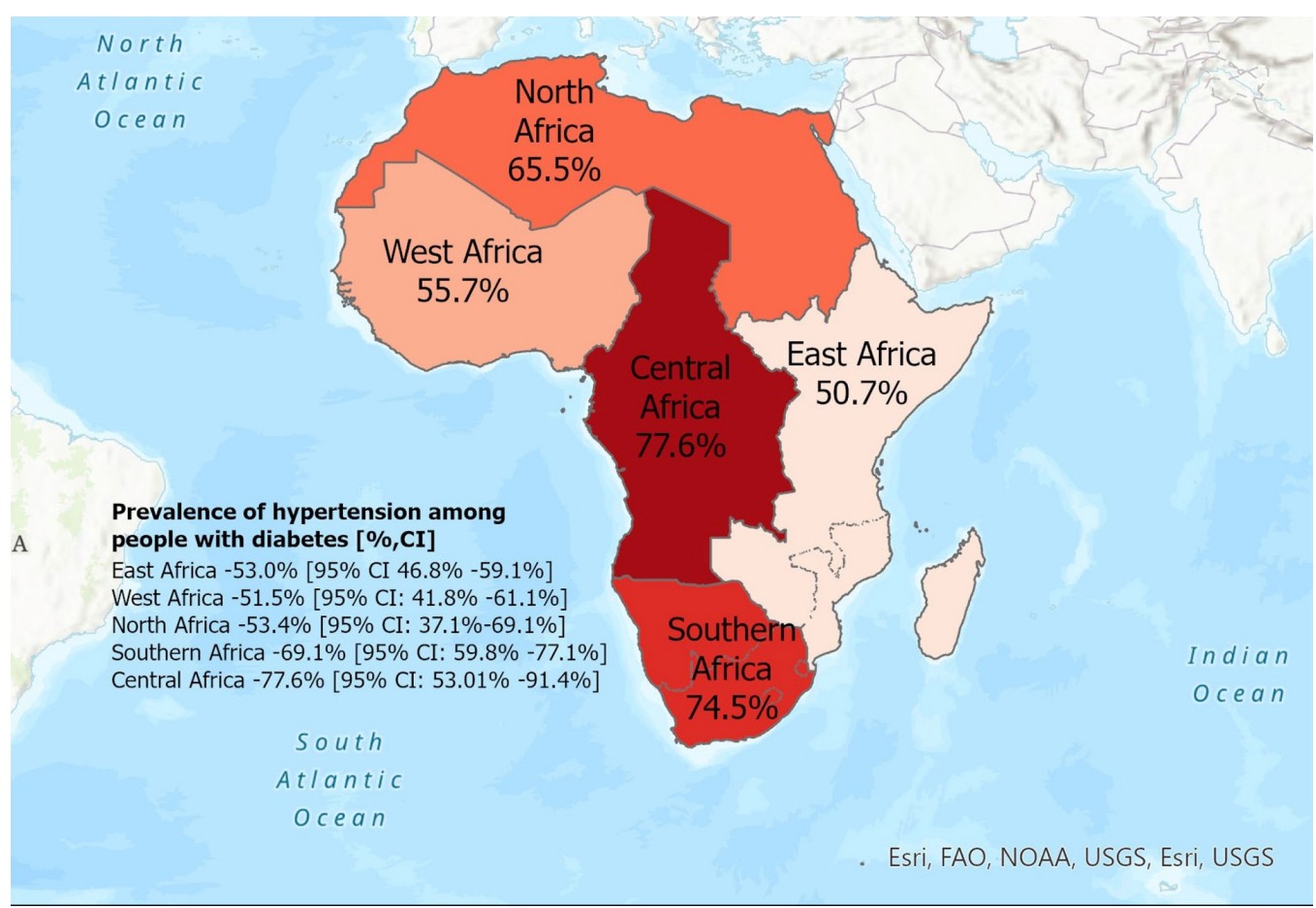

**Fig 3.**

category [31]. The duration of diabetes is also associated with an increasing risk of developing hypertension. In three studies, being male increases the odds of developing hypertension among people diagnosed with diabetes. With regards to the residence, Akalu et al. (2020) and Awadalla et al. (2017) reported that people diagnosed with diabetes living in urban residence have higher chances of developing hypertension.

## Discussion

We comprehensively reviewed the available literature and conducted a meta-analysis on the prevalence of hypertension among people with diabetes in Africa. The pooled prevalence of hypertension was 58.1%, with high statistical heterogeneity across studies. The prevalence varied across African sub-regions, with Central Africa having the highest prevalence of (77.6%) with West Africa recording the lowest (51.5%). This finding is consistent with a systematic review by Colosia et al. who reported a hypertension prevalence between 38.5–80% among people with diabetes globally [63]. Comparably, while the overall prevalence of hypertension among people with diabetes is slightly higher than the prevalence (55.2%) in the general population in Africa [64], the sub-regional analysis was much higher in the North, Central, and South African sub-regions which can be attributed to the differences in socio-cultural practices, food insecurity, differences in access to healthcare and socio-economic conditions in the

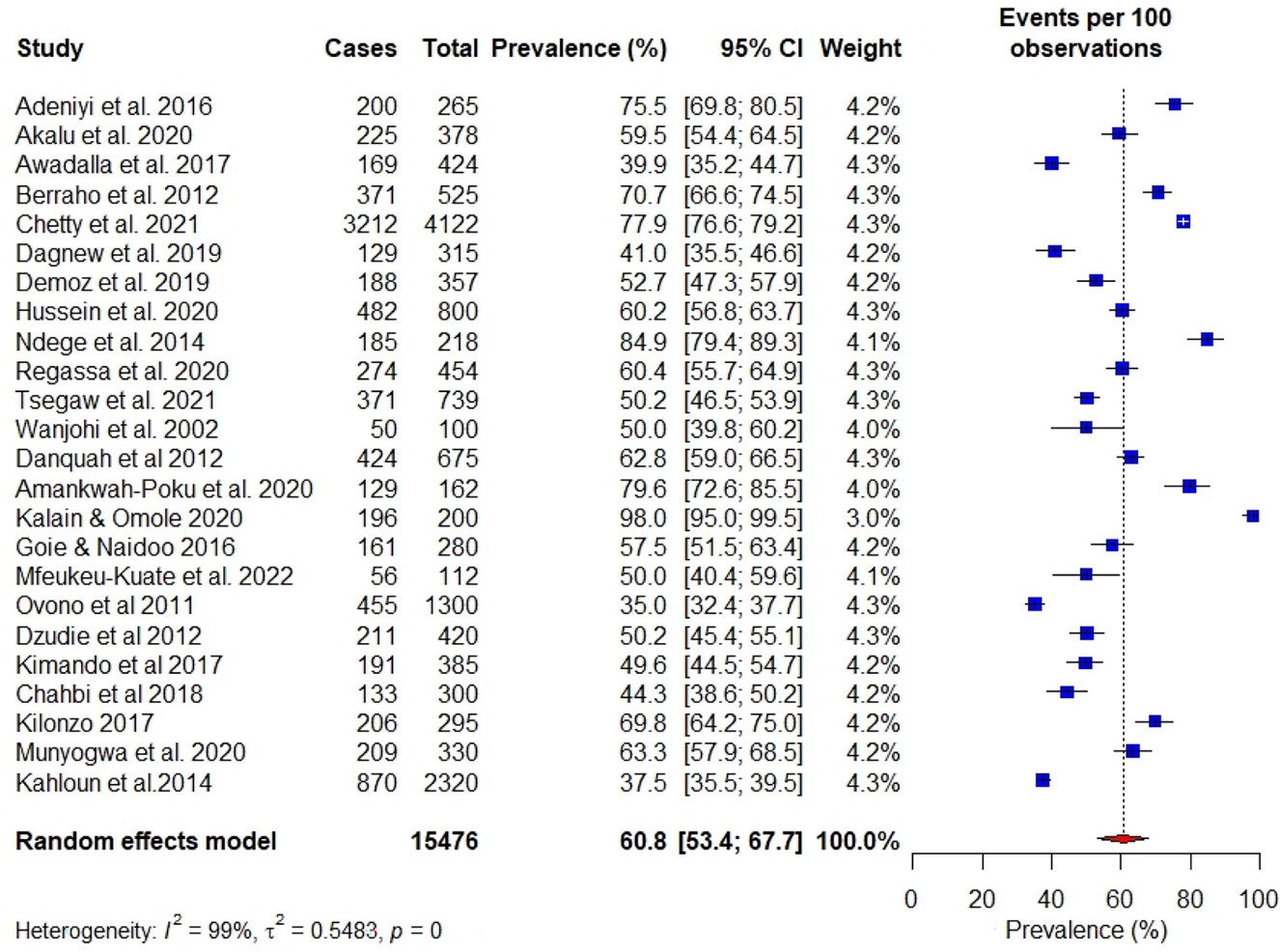

**Fig 4.**

sub-region. Furthermore, this high heterogeneity can be explained in terms of the different diagnostic criteria for the diagnosis of hypertension and diabetes in the region.

This review provides evidence of the increasing risk of hypertension among people with diabetes in Africa. For instance, a review conducted in Ethiopia reported a hypertension prevalence of 55% among patients with diabetes, which aligns with the 53% prevalence recorded in this review for the East Africa sub-region [65]. While this highlights the need for further research to explore the physiological mechanism between the risk of hypertension among people with T2DM in the region, the few studies identified from North and Central Africa regions in this review point to the limited research on hypertension among people with diabetes in these African sub-regions.

Among people diagnosed with T2DM, the prevalence of hypertension was 60.8%, which was higher than the overall prevalence among people with both T1DM and T2DM combined. The South African region recorded the highest prevalence of (92%) and its lowest at 57.5%. This finding can be explained in the context of urbanization, increasing prevalence of overweight and obesity in the Southern African region compared to East and West African regions. Given that obesity is a significant risk factor for hypertension, it was not surprising to find that

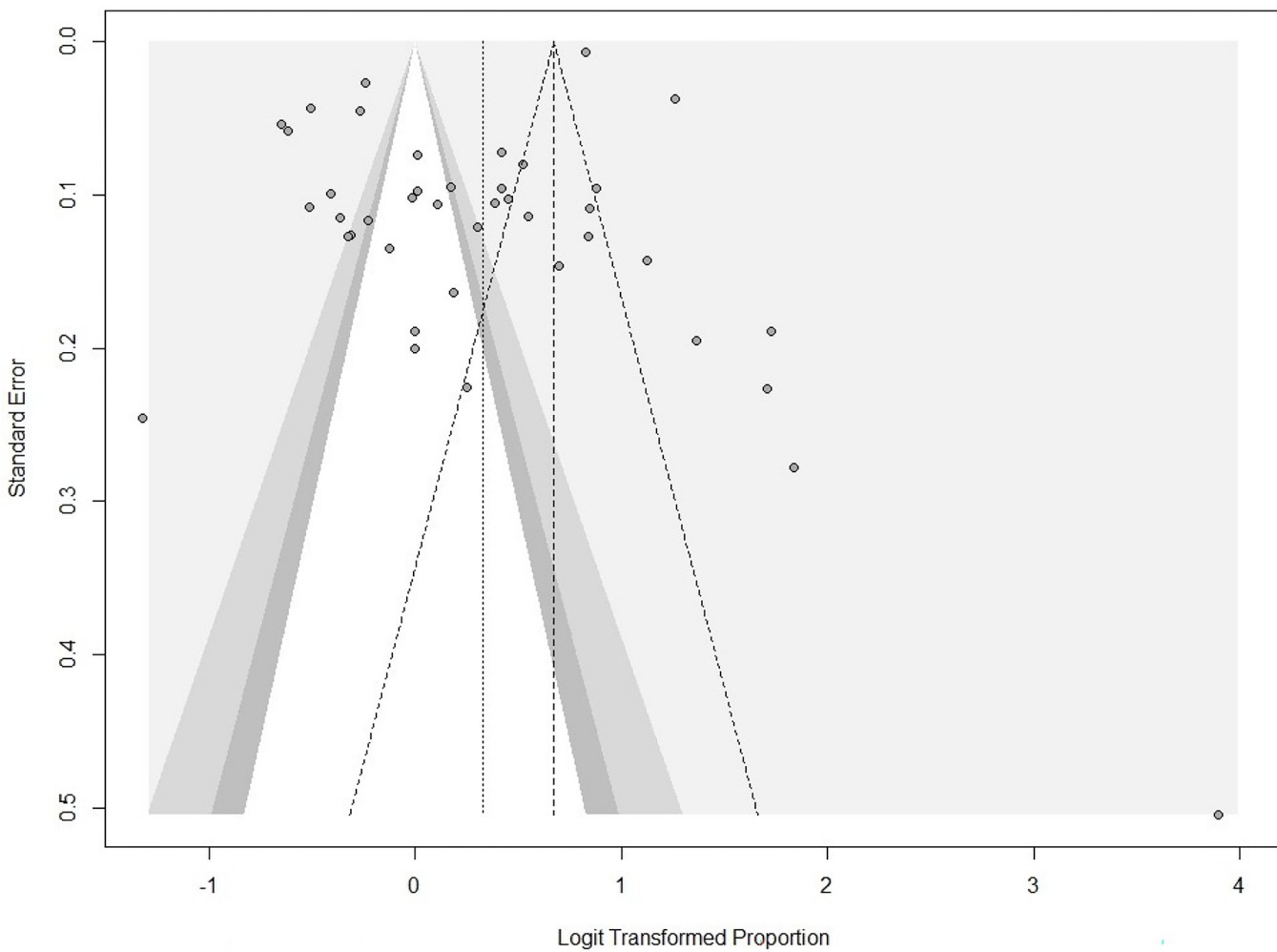

**Fig 5.**

African regions that reported the highest prevalence of obesity also recorded the highest prevalence of hypertension among its diabetes population.

Additional analysis conducted identified age, long duration of diabetes, male sex, and living in urban areas as factors associated with the developing hypertension-diabetes co-morbidity. This finding corroborates the findings of the largest United Kingdom Prospective Diabetes Study (UKPDS) in 1998 which reported an association between age, bigger waist/hip ratio, sedentary lifestyle with diabetes development, and further cardiovascular risk [66]. Moreover, other studies have shown that time since diabetes diagnosis is positively and independently associated with the risk of developing macrovascular and macrovascular complications among people with diabetes.

## Strengths and limitations

Previous reviews have focused on hypertension or diabetes burden in infectious diseases such as HIV/AIDS and Tuberculosis in Africa. However, this study is the first review to estimate the prevalence of hypertension among people diagnosed with diabetes in Africa. Additionally, this review captured the prevalence among people with T2DM and T1DM. However, the review

needs to be considered in the context of some limitations. Only studies published in English were included and that may cause language bias, especially as some African countries publish in other languages like French. More specifically, countries in central Africa were likely to be under-represented which could affect the prevalence estimates. Nonetheless, this review provides a snapshot of the prevalence of hypertension among people with diabetes in Africa. Additionally, we searched only three databases and may have not captured all the studies conducted in the region about hypertension/diabetes comorbidity. Lastly, there was a high statistical heterogeneity across studies even after accounting for regional variations. The high heterogeneity in our meta-analysis may limit the generalizability of our pooled prevalence estimate and should be considered when interpreting the overall findings.

## Implications for policy and practice

The high burden of hypertension among people with diabetes in Africa presents significant implications for health policy and research. Firstly, health systems should be strengthened to meet the needs of people living with hypertension/diabetes comorbidity through the adoption of an integrated model of care that provides an opportunity to cost-effectively manage both hypertension and diabetes without further strain on resources and healthcare providers. Community-based interventions are required to increase awareness of hypertension and diabetes in Africa. The evidence from this review also supports the adoption of a team-based care policy, and further research is required to examine its feasibility in the African region.

## Supporting information

**S1 Checklist. PRISMA 2009 checklist.**
(PDF)

**S1 Table. Search strategy.**
(PDF)

**S2 Table. Assessment of risk of bias.**
(PDF)

**S1 Fig. Prevalence of hypertension among people with diabetes in West Africa: Systematic review, and meta-analysis.**
(TIF)

**S2 Fig. Prevalence of hypertension among people with diabetes in West Africa: Systematic review, and meta-analysis.**
(TIF)

**S3 Fig. Prevalence of hypertension among people with diabetes in North Africa: Systematic review, and meta-analysis.**
(TIF)

**S4 Fig. Prevalence of hypertension among people with diabetes in South Africa: Systematic review, and Meta-analysis.**
(TIF)

**S5 Fig. Prevalence of hypertension among people with diabetes in Central Africa: Systematic review, and meta-analysis.**
(TIF)

**S6 Fig. Forest plot of studies showing studies with a high-quality score of 4 or better.**
(TIF)

**S7 Fig. Forest plot of high-quality studies included only participants with type 2 diabetes.**
(TIF)

## Acknowledgments

We thank Emily Hoppe a Ph.D. candidate at the Johns Hopkins University School of Nursing for providing support on using the ArcGIS platform to visually present the map prevalence of hypertension on the African map.

## Author Contributions

**Conceptualization:** Thomas Hinneh, Samuel Akyirem, Irene Fosuhemaa Bossman, Victor Lambongang, Patriot Ofori-Aning, Oluwabunmi Ogungbe.

**Data curation:** Thomas Hinneh, Samuel Akyirem, Irene Fosuhemaa Bossman, Victor Lambongang, Patriot Ofori-Aning, Oluwabunmi Ogungbe.

**Formal analysis:** Thomas Hinneh, Samuel Akyirem.

**Investigation:** Victor Lambongang.

**Methodology:** Thomas Hinneh, Samuel Akyirem.

**Supervision:** Oluwabunmi Ogungbe, Yvonne Commodore Mensah.

**Visualization:** Thomas Hinneh.

**Writing – original draft:** Thomas Hinneh, Samuel Akyirem, Irene Fosuhemaa Bossman, Oluwabunmi Ogungbe.

**Writing – review & editing:** Thomas Hinneh, Samuel Akyirem, Irene Fosuhemaa Bossman, Patriot Ofori-Aning, Oluwabunmi Ogungbe, Yvonne Commodore Mensah.

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
