## [Decision Letter · Decision Letter 0]

26 Sep 2023

PGPH-D-23-00718

Regional Prevalence of Hypertension Among People Diagnosed with Diabetes in Africa, A Systematic Review and Meta-analysis

Dear Dr. Hinneh,

Thank you for submitting your manuscript to PLOS Global Public Health. After careful consideration, we feel that it has merit but does not fully meet PLOS Global Public Health’s publication criteria as it currently stands. Therefore, we invite you to submit a revised version of the manuscript that addresses the points raised during the review process.

Dear Dr. Hinneh,

Please see the reviewer comments before we can proceed with your submission.

Kind regards,

Leonor Guariguata

We look forward to receiving your revised manuscript.

Kind regards,

Leonor Guariguata, MPH, PhD

Academic Editor

Journal Requirements:

1. Some material included in your submission may be copyrighted. According to PLOS’s copyright policy, authors who use figures or other material (e.g., graphics, clipart, maps) from another author or copyright holder must demonstrate or obtain permission to publish this material under the Creative Commons Attribution 4.0 International (CC BY 4.0) License used by PLOS journals. Please closely review the details of PLOS’s copyright requirements here: PLOS Licenses and Copyright. If you need to request permissions from a copyright holder, you may use PLOS's Copyright Content Permission form.

Potential Copyright Issues:

Figure 2: please (a) provide a direct link to the base layer of the map (i.e., the country or region border shape) and ensure this is also included in the figure legend; and (b) provide a link to the terms of use / license information for the base layer image or shapefile. We cannot publish proprietary or copyrighted maps (e.g. Google Maps, Mapquest) and the terms of use for your map base layer must be compatible with our CC-BY 4.0 license. 

Additional Editor Comments (if provided):

Reviewers' comments:

Reviewer's Responses to Questions

**Comments to the Author**

1. Does this manuscript meet PLOS Global Public Health’s publication criteria? Is the manuscript technically sound, and do the data support the conclusions? The manuscript must describe methodologically and ethically rigorous research with conclusions that are appropriately drawn based on the data presented.

Reviewer #1: No

Reviewer #2: Partly

2. Has the statistical analysis been performed appropriately and rigorously?

Reviewer #1: No

Reviewer #2: No

3. Have the authors made all data underlying the findings in their manuscript fully available (please refer to the Data Availability Statement at the start of the manuscript PDF file)?

Reviewer #1: Yes

Reviewer #2: Yes

4. Is the manuscript presented in an intelligible fashion and written in standard English?

Reviewer #1: Yes

Reviewer #2: Yes

5. Review Comments to the Author

Reviewer #1: The authors have attempted to estimate the burden of non-communicable diseases in Africa, which is an important problem in the region. However, I have some major concerns with the methods used to achieve this aim. The following are my comments.

1. The study rationale for estimating the prevalence of HTN among diabetes is neither clear from the introduction nor a necessary one. I believe it is more important to estimate the prevalence of joint occurrence of HTN and DM in the population. So instead of using Diabetics as the study population, I would suggest to have the entire population as the study population and estimate the prevalence of co-occurrence of HTN and DM in the general population. The search strategy given by the authors may be sufficient for this objective.

2. For prevalence studies, the standard is to include representative surveys conducted in the community. It is not clear whether the authors included only such studies or whether they included hospital or facility-based studies as well.

3. For LMICs, Google Scholar is an important database to search for published/unpublished literature. I suggest to run a search in Google Scholar for additional studies.

4. The eligibility criteria do not mention types of study designs that will be included in the review.

5. MMAT is not the correct tool for assessing the quality of cross-sectional surveys in SR, if that was the study design included in the review.

6. In statistical methods, the authors have mentioned GLM with logit transformation. It is not clear why this methodology was used. For a simple prevalence meta-analysis, it is not required to do a logit transformation.

7. There is no mention of the high level of heterogeneity in the pooled estimate.

Minor comments

1. Abstract is not written as per the PRISMA guideline. [https://doi.org/10.1371/journal.pmed.1001419]

2. Mean age cannot be calculated in this manner in a SRMA. A simple average of extracted mean ages from each study is not statistically correct.

3. It is not possible in this study to determine the risk factors for HTN or DM and how the p-values were calculated for this is not written.

4. In the introduction, the authors have used the word prevalence and proceed to numbers in million. Prevalence has to be given as a proportion or percentage per unit of population or instead the word prevalence may be avoided.

5. Table 1 is not cited in the main text. Main tables should not be provided in the supplement but in the main text itself.

Reviewer #2: I thank the authors for their attempt to capture the burden of hypertension among persons with diabetes in this population given the impact the double burden is likely to have on policy. By their own background, it is clear that it is fairly well known that the burden is high in the diabetes subpopulation but the attempt to quantify it is noted.

I recommend the authors make some changes and clarify a few issues before this publication is accepted. The limited use of databases, limited language and limited grey literature search makes this review not as comprehensive as it could be and this should be noted in limitations.

The background is well written and provides rationale for the study. There are areas of repeated text-

• Lines 117-120 and

• Lines 120-121 and Lines 124-126 are paraphrased but essentially the same.

I could find no reference in the body of the results to Table 1.

For clarification

1. Kindly explain what the “population, outcome, region” strategy is. It is quoted as if a standard. Can you provide a reference for where it is derived or explain how the authors derived the strategy if it de novo.

2. Authors state that the Hypertension cut off was specified as 140/90 and 130/80. Can you reference the guidelines used to make this specification? Under what circumstances did you accept the various definitions. The latest guidelines of the 2022 ADA recommend that BP should be con- trolled to <140/90 mm Hg in patients with diabetes mellitus, but <130/80 mm Hg if there is a higher CV risk with existing ASCVD or 10-year ASCVD risk of ≥15%. I suspect that the authors used older guidelines and should provide reference and clarity on the circumstances (comorbidities) under which they accepted the various cut off points.

3. Line 165-166: Kindly indicate where it was applicable to use MESH terms? On which terms in PUBMED as this used? Was it used on some and not others since the authors say, “where applicable”

4. The search strategy stating only “Africa” or African countries seems limiting to me. Was NESH used here?

5. Why was the MMAT used for risk of bias assessment? Given that the study focused heavily on prevalence which would be primarily quantitative studies can authors justify the use of this tool which is usually for mixed methods?

6. Line 222-223 states: “32 (78%) of the studies utilized a cut-off point of >140/90 mm Hg for hypertension diagnosis, some 3 (7.3%) used >130/80 mm Hg as the cut-off.” What did the other

studies use as their cut-off?

7. Can the authors explain what was done when the strategy was updated? I understand that the search was repeated but I am not clear on how this was matched against what already existed? Was the title and abstract screening repeated? Was there a duplicate comparison between the old and new list? Using what software?

Major changes requested

1. Even though the authors mention in one sentence in the abstract the use of I2 to assess heterogeneity, no such assessment is described in the actual text of the study. Please describe how heterogeneity was assessed. How was I2 used? What were the cut off values? Did you also conduct a qualitative assessment of heterogeneity? Please describe more fully in the methods of main text.

2. Please describe the results of the risk of bias assessment. This is a critical component of systematic reviews and should not be a footnote in the methods. How did the risk of bias assessment affect the use of the studies? Did the scores influence the authors decisions on which studies should be in the meta-analysis?

For example, given the high heterogeneity, were attempts made to do meta-analysis on studies with only the best quality assessment scores? I highly recommend that the authors conduct meta-analyses only on studies with risk of bias of 4/5 to determine the prevalence in these low risk studies. This would be a suitable form of sensitivity analysis especially as the I2 scores are so high.

3. Since age emerged qualitatively as a major factor in hypertension prevalence, consider also doing meta-analysis in two major age groups, again it may help reduce heterogeneity and provide a clearer picture of prevalence that could inform health care access planning.

4. I caution how the authors interpret their findings. They state that their paper shows evidence of ever increasing prevalence of hypertension and given the lack of trending data, it does not show this. There was no previous systematic review in Africa for comparison so I recommend perhaps simply stating that the burden is high as opposed to increasing

Limitations

The lack of grey literature search and limited to only three databases is a weakness of the study that should be acknowledged in the limitations section. Especially given that the authors actually found publication bias.

6. PLOS authors have the option to publish the peer review history of their article (what does this mean?). If published, this will include your full peer review and any attached files.

**Do you want your identity to be public for this peer review?** For information about this choice, including consent withdrawal, please see our Privacy Policy.

Reviewer #1: No

Reviewer #2: No

---

## [Decision Letter · Decision Letter 1]

7 Nov 2023

Regional Prevalence of Hypertension Among People Diagnosed with Diabetes in Africa, A Systematic Review and Meta-analysis

PGPH-D-23-00718R1

Dear Mr Hinneh,

We are pleased to inform you that your manuscript 'Regional Prevalence of Hypertension Among People Diagnosed with Diabetes in Africa, A Systematic Review and Meta-analysis' has been provisionally accepted for publication in PLOS Global Public Health.

Best regards,

Leonor Guariguata, MPH, PhD

Academic Editor

Reviewer Comments (if any, and for reference):

Reviewer's Responses to Questions

**Comments to the Author**

1. If the authors have adequately addressed your comments raised in a previous round of review and you feel that this manuscript is now acceptable for publication, you may indicate that here to bypass the “Comments to the Author” section, enter your conflict of interest statement in the “Confidential to Editor” section, and submit your "Accept" recommendation.

Reviewer #1: All comments have been addressed

Reviewer #2: All comments have been addressed

2. Does this manuscript meet PLOS Global Public Health’s publication criteria? Is the manuscript technically sound, and do the data support the conclusions? The manuscript must describe methodologically and ethically rigorous research with conclusions that are appropriately drawn based on the data presented.

Reviewer #1: Yes

Reviewer #2: Yes

3. Has the statistical analysis been performed appropriately and rigorously?

Reviewer #1: Yes

Reviewer #2: Yes

4. Have the authors made all data underlying the findings in their manuscript fully available (please refer to the Data Availability Statement at the start of the manuscript PDF file)?

Reviewer #1: Yes

Reviewer #2: Yes

5. Is the manuscript presented in an intelligible fashion and written in standard English?

Reviewer #1: Yes

Reviewer #2: Yes

6. Review Comments to the Author

Reviewer #1: Responses were satisfactory

Reviewer #2: Thank you for addressing comments made.

7. PLOS authors have the option to publish the peer review history of their article (what does this mean?). If published, this will include your full peer review and any attached files.

**Do you want your identity to be public for this peer review?** For information about this choice, including consent withdrawal, please see our Privacy Policy.

Reviewer #1: No

Reviewer #2: **Yes: **Natasha Sobers
